# Navigating the Post-API Dilemma

## Search Engine Results Pages Present a Biased View of Social Media Data

Double Blind

## ABSTRACT

Recent decisions to discontinue access to social media APIs are beginning to have detrimental effects on Internet research and the field of computational social science as a whole. This lack of access to data has been dubbed the *Post-API* era of Internet research. Fortunately, popular search engines have the means to crawl, capture, and surface social media data on their Search Engine Results Pages (SERP) if provided the proper search query, and may provide a solution to this dilemma. In the present work we ask: does SERP provide a complete and unbiased sample of social media data? Is SERP a viable alternative to direct API-access? To answer this question, we perform a comparative analysis between (Google) SERP results and nonsampled data from Reddit and Twitter/X. We find that SERP results are highly biased in favor of popular posts; against political, pornographic, and vulgar posts; are more positive in their sentiment; and have large topical gaps. Overall, we conclude that SERP is not a viable alternative to social media API access.

## CCS CONCEPTS

• **Information systems** → **Collaborative and social computing systems and tools**; **Web searching and information discovery**; • **Human-centered computing** → **Collaborative and social computing**.

## KEYWORDS

Social Media, Data Access, Bias, Search

**ACM Reference Format:**
Double Blind. 2018. Navigating the Post-API Dilemma: Search Engine Results Pages Present a Biased View of Social Media Data. In *Proceedings of Make sure to enter the correct conference title from your rights confirmation emai (Conference acronym 'XX)*. ACM, New York, NY, USA, 9 pages. https://doi.org/XXXXXXX.XXXXXXX

> This paper contains material that may not be suitable for all audiences. Reader discretion is advised.

## 1 INTRODUCTION

In February 2023, Twitter/X announced its plan to discontinue free access to its API-services. Shortly thereafter, the prominent Web site and discussion forum Reddit announced that it would take a similar action and likewise discontinue free API-access to its site data. In an interview with the New York Times, Steve Huffman, the

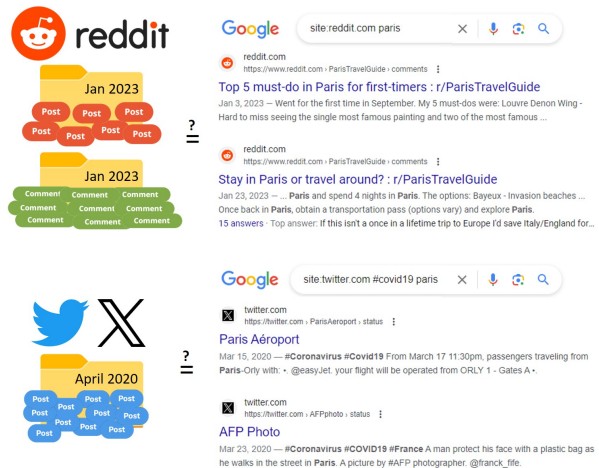

**Figure 1: Problem definition. We ask: does Google-SERP provide an unbiased sample of social media data? Is Google-SERP a valid replacement for social media APIs?**

CEO of Reddit, explained his rationale stating, "The Reddit corpus of data is really valuable, but we don't need to give all of that value to some of the largest companies in the world for free [13]." The discontinuation of API access on both Twitter/X and Reddit led to a backlash from developers, especially those of the third-party applications, which rely heavily on API-access from these sites. Many of these third-party applications and their companies have since shuttered their operations. A similar effect has been felt among researchers and academics who rely on access to data for scholarship in countless areas of study.

For example, scholars have been using access to Reddit's API-service almost since its founding in 2006. This data has led to several studies, especially in the field of discourse [5], computational journalism [31], and computational linguistics [2, 15, 21, 36] to name a few. This is true even moreso for Twitter/X, which has seen numerous studies on follower networks [18, 38], event detection [35, 37], and coordinated influence campaigns[14, 27]. Without access to data, this type of scholarship will be difficult or impossible. We call this the *Post-API Dilemma* and in this paper we begin to ask the question: How shall scientists continue their work without access to this data?

*Web Scraping.* One way to continue this work is through the clever use of Web scrapers, which are computer programs that pretend to be regular social media users clicking and scrolling a rendered newsfeed, but are in fact sophisticated systems that collect data for later analysis. These programs are wildly inefficient compared to the firehose of information that can be gleaned from API-access. As a result, Web scrapers will not be able to sufficiently replace the lack of API-access, even in the best case.

*Search Engine Results Pages.* Despite the API cutoff for most developers and researchers, many of the popular search engines still have direct API-access or are permitted (and have the resources) to scrape social media systems enmass. Because these search engines also provide API-access to their Search Engine Results Pages (SERP) it may suffice to collect data from social media systems indirectly by issuing properly-formed queries to a SERP-API. Guidelines for SERP-API access vary across providers, but, in essence, SERP-APIs take in a query-string akin to the advanced search page of most search engines and return the search results in a desired format (*e.g.*, json, xml).

The availability of indirect access to social media data via SERP leads to several important questions. What is the coverage of the SERP compared to direct access to social media system? Does SERP provide a random sample of the social media data? Or are the results biased in any meaningful way? If they are biased, can we characterize the nature of the bias? It is critical that we understand the answers to these questions if we are to use SERP as an alternative, indirect means of social media data collection.

In the present work, we make an initial attempt to answer these questions by comparing known complete snapshots of social media datasets against results from SERP responses. We focus specifically on Reddit and Twitter/X because these two social media site were the two most high-profile sites to revoke their API-access; and we compare their data against results from the Google search engine accessed by the API service ScaleSERP[1].

*Preliminaries.* The wide-availability of social media data has been an enormous benefit in the study of online human behavior [25, 40]. Due to the widespread use of this data, it is important to understand if the systems that deliver this data are indeed providing an unfiltered view of the behavior that the researcher seeks to investigate. For example, the methodology used by Pushshift to collect Reddit data is often challenged by collection delays; even though these collection delays were sometimes only a few minutes, posts and comments that were deleted during the intervening period could be missed thereby affecting important research into content moderation [11] or self-moderation [28]. Likewise, the Twitter gardenhose API, which (prior to 2023) returned at most a 1% sample of Twitter data and which has served as the go-to data source for computational social scientists over the past decade, has been shown to be a highly biased sample of the Twitter corpus [24]. For example, Morestatter *et al.* found that more than 90% of the geotagged Tweets were present in the 1% garden-hose sample [24].

The law of large numbers tell us that the mean of a sample converges to the mean of the population as the sample-size increases. This gives hope that a large enough sample will provide enough information to draw reasonable conclusions. What remains, then, is to simply collect enough samples to draw statistically informed conclusions. Unfortunately, this statistical *law* only applies when the sample set is selected *randomly*. Although the calculation of search engine results is a closely guarded secret, it is safe to assume that the results-sample provided by search engines are not random.

The goal of the present work is to characterize the differences between these corpora so that scholars who make use of these tools

do so with these limitations in mind. Specifically we characterize the differences and biases along the following dimensions of analysis:

(1) Popularity Analysis. We first determine any relationship between user or post popularity (in terms of Twitter/X followers or Reddit score) and the results and rank returned by SERP.
(2) Token-level. We employ an information-theoretic text analysis method called Rank Turbulence Divergence (RTD) to investigate which terms are more likely to appear in one corpus but not in the other, and vice versa.
(3) Sentiment-level. We employ state-of-the-art NLP tools to compare the aggregate sentiment (positive, neutral, negative) between the corpora.
(4) Topic-level. We perform a topic analysis over unigrams and characterize any topical gaps that occur between the corpora.

*Findings.* In summary, SERP results of social media platforms were highly biased in several ways. Therefore, SERP is not a replacement for direct API access from the social media platforms.

More specifically, along the four dimensions of analysis we find: (1) SERP yields posts that are significantly more popular (in terms of number of followers on Twitter/X and score on Reddit) than the average Reddit post or active Twitter user. However, the rank of the post from SERP is not correlated with popularity, *i.e.*, the first result is not more popular than the 100th result. (2) Compared to SERP, the nonsampled social media data is substantially and statistically different; for example, nonsampled social media data had a much larger frequency of politically-oriented language compared to results from SERP. (3) Compared to SERP, the nonsampled social media posts had statistically more negative sentiment than SERP results. (4) We found large coverage gaps in the semantic/topic space produced by the SERP compared to the nonsampled social media data.

The details of the methodology are important and are described in Section 2. Then we describe the precise analysis methodology and their results for each of the four dimensions of analysis in Sections 3, 4, 5, and 6 before we conclude with a discussion on threats to validity and future work.

## 2 DATA COLLECTION METHODOLOGY

To confidently study any biases in SERP, it is important to obtain strong unbiased social media datasets from which to compare. Specifically, we investigate Reddit and Twitter/X. In both cases we are confident that the collected samples are nearly complete within a specific time window.

### 2.1 Reddit Data

We collected data from Pushshift[2] up until March of 2023. This data nearly complete, but may lack posts and comments that were either identified as spam by Reddit or deleted, edited, or removed by the moderation team or by the user before the Pushshift data collection service was able to collect the data, or was otherwise inaccessible by virtue of the post or comment being in a quarantine subreddit or otherwise. Nevertheless, it is likely that this dataset contains almost all the social media content that was visible to a regular

---

[1]http://scaleserp.com

[2]http://pushshift.io

user of Reddit. The number of up-/downvotes, awards, flairs, and other metadata associated with a post changes regularly; these changes are not reflected in this dataset. However, the present work mostly considers the text-content of the posts and comments so the ever-changing metadata is not relevant.

We focus our investigation on the posts and comments submitted to Reddit between January 1, 2023 and January 31, 2023. This timeframe was chosen because it is recent, complete, and large. In total this subset contains 36,090,931 posts, and 253,577,506 comments. We tokenized this dataset, removed any tokens that contained non-alphabetic characters, ranked each token according to its document frequency, and selected 1000 keywords by stratified sampling. Stratified sampling is used to ensure that the set of keywords are uniformly distributed from common to rare words, *i.e.*, they are not dominated by words of one type or another.

## 2.2 Twitter/X

Obtaining a complete set of Twitter/X data is difficult, even for a single day [30]. To make matters worse, new restrictions limit the sharing of data to only the identifiers, which do not contain the content of the post. Fortunately, there do exist Twitter/X datasets that are nearly complete for some subset of the platform. Specifically, we used a dataset of Tweets related to the COVID-19 pandemic, which was collected for one month starting on March 29, 2020, and ending on April 30, 2020 [1]. This dataset is considered to be a nearly-complete set of tweets that contain one of seven different hashtags, (*e.g.*, #coronavirus, #coronavirusoutbreak, #covid19) for a total of 14,607,013 tweets during the time period.

## 2.3 SERP Data

Search engines like Bing and Google have the infrastructure to collect social media data at a massive scale. Researchers who rely on data access have been turning to services that provide relatively inexpensive SERP-API access.

It is infeasible to simply ask SERP for a list of all tweets. So we used the SERP-API system to query Google with each of the 1000 random keywords extracted from the Reddit dataset.

Comparing against the Reddit dataset required each query to be of the form: `site:reddit.com {keyword}`. We found that the majority of queries were limited to 100 results each, so we repeated each query setting the date restriction for one day-at-a-time for each day in January 2023 thereby matching the timeframe from Reddit-dataset. All other options were kept at their Google-defaults except safe-search, which we disabled. Furthermore, the ScaleSERP service notes that they use thousands of proxies distributed throughout the world, so the results presented in the current study should not be biased by geographical region.

Comparing against the Twitter/X dataset used a similar methodology, except the queries needed to also include one of the hashtags used to obtain the Twitter data in order to maintain a fair comparison. The Twitter query to SERP was of the form: `site:twitter.com {hashtag} {keyword}` for each keyword. Because many tweets contained more than one of the relevant hashtags we randomly sampled a single covid-hashtag for each keyword. We were again careful to match the dates from the Twitter/X dataset. Like in the Reddit-SERP

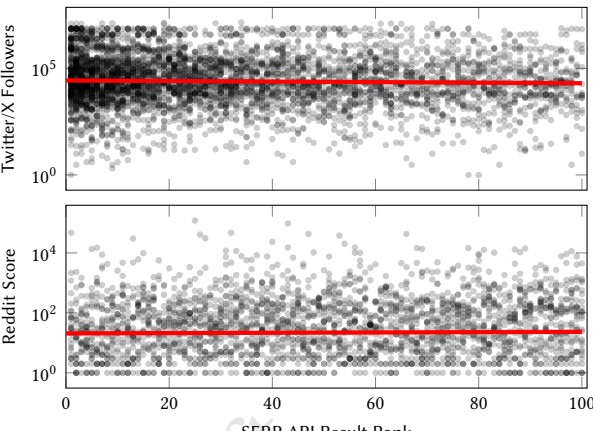

**Figure 2: Twitter/X user followers (top) and Reddit post score (bottom) as a function of the SERP result rank. Red lines show exponential regression. We find almost no correlation between user or post popularity and rank from SERP.**

methodology, all other options were kept at their Google-defaults except safe-search, which we again disabled.

## 2.4 Data Models

Relative to the enormous size of the nearly-complete Reddit and Twitter/X datasets, the results gathered from SERP were surprisingly small. In total SERP gathered 1,296,958 results from Reddit and 70,018 tweets from Twitter/X. Note that the results for Reddit are typically links to entire comment threads, but SERP results for Twitter/X are typically link to a single Tweet.

Results from SERP are indexed for later comparison. Data for Reddit includes the post/comment-id, userid, score, post-title, post-text, and all comments on the post. Results for Twitter/X only contain the userid, Tweet-id, and the Tweet-content. With this data, it is possible to perform a comparison of the data gleaned from SERP against the data known to exist on Reddit and Twitter/X.

## 3 POPULARITY ANALYSIS

We begin with an analysis that characterizes any relationship between the popularity of a user or the score of a post and its typical rank from SERP. Search engines are well known to promote highly authoritative sources [33], and this may (or may not) be true for the ranked results for social media searches. We expect that high-scoring Reddit posts and Tweets from highly influential Twitter/X users will dominate the SERP results, thereby introducing a bias in the overall search results.

To characterize any ranking biased induced by SERP, we plotted the number of followers for the user who posted each returned Tweet as a function of the rank in SERP. Most SERP queries returned slightly more than 100 results, so we plot the relationship between the number of followers of the Tweeting user and its rank at the top of Fig. 2. Popularity on Reddit is different than on Twitter/X in that users rarely have any following at all; instead, post-score is typically used as the signal for popularity. Therefore, the bottom plot in Fig. 2 shows the relationship between the score of a post

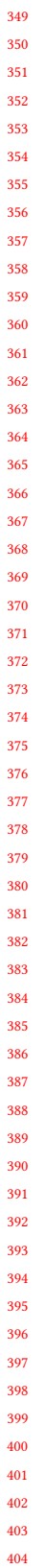

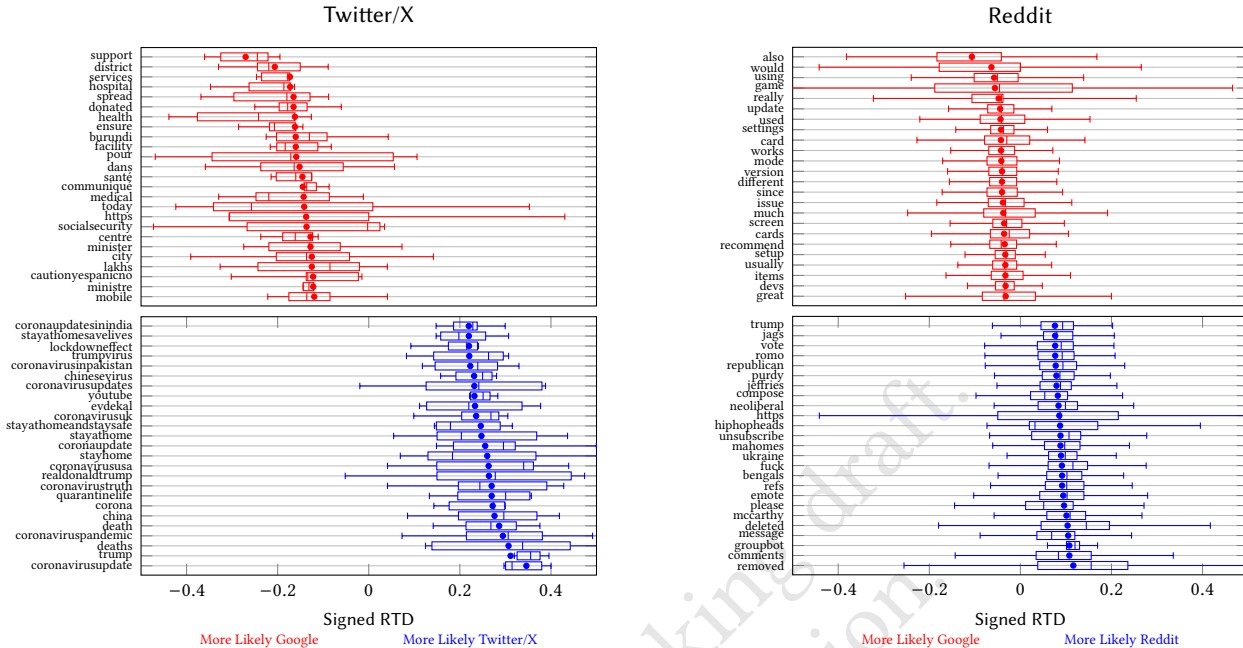

**Figure 3: Signed Rank Turbulence Divergence (RTD) for the most divergent terms comparing results from SERP against Twitter/X (on left) and against Reddit (on right). Terms that are more likely to appear in SERP results are listed on top (red). Terms that are more likely to appear in the nonsampled social media data are listed on the bottom (blue). We find that the social media posts returned by SERP are far more likely to contain innocuous terms compared to the nonsampled social media data.**

and its rank. The red-line in both plots shows the result of an exponential regression. In both cases, the line was almost perfectly flat. Spearman correlation test showed almost no correlation between a Twitter/X user's follower count and its rank from SERP ($R^2$=0.002); a similar analysis found almost no correlation between the score of the Reddit post and its rank from SERP ($R^2$=0.001).

In short, contrary to our expectation, we do not find evidence that Twitter/X user popularity nor Reddit scores affect the SERP results. However, it is worth noting that the mean and median post-score in our Reddit dataset was 48.97 and 1.0 respectively compared to 550.69 and 21.0 from SERP; the mean and median number of followers in our Twitter/X dataset was 63,250.16 and 873.0 respectively compared to 544,934.63 and 21,547.0 from SERP. So, although there was no rank-correlation it does appear that SERP does return scores that are statistically significantly higher than the typical Reddit post (MannWhitney $\rho$=0.259 p<0.001) and the typical active Twitter/X user (MannWhitney $\rho$=0.194 p<0.001).

## 4 TOKEN-BASED COMPARISON

Next we look to identify token-level discrepancies between the datasets. Typical token-based analysis takes the view that the text-datasets can be represented as a bag-of-words. Then, any number of statistical analysis can be employed to compare these categorical distributions [6, 7]. But these traditional distances are difficult to interpret when the data is Zipfian [10], as most text-data is [23].

Recently, the Rank Turbulence Divergence (RTD) metric was introduced as an illuminating and information-theoretic measure of the difference between two text-corpora [8]. We employ this measure to identify any token-level sample bias from SERP.

Formally, let R1 and R2 be two word distributions ranked from most common to least common. To start, the RTD calculates the element-wise divergence as follows:

$$\left| \frac{1}{r_{\tau,1}} - \frac{1}{r_{\tau,2}} \right| \tag{1}$$

where $\tau$ represents a token and $r_{\tau,1}$ and $r_{\tau,2}$ denote its ranks within R1 and R2, respectively. Because Eq. 1 introduces bias towards higher ranks, the authors introduce a control parameter $\alpha$ as:

$$\left| \frac{1}{[r_{\tau,1}]^\alpha} - \frac{1}{[r_{\tau,2}]^\alpha} \right|^{\frac{1}{\alpha+1}}. \tag{2}$$

For each term present in the combined domain of R1 and R2, we compute their divergence using Eq. 2. We take care to preserve the sign, which, when negative, signifies that an element holds a higher rank in R1, while a positive sign indicates the opposite. In the present work we use $\alpha = \frac{1}{3}$ empirically.

The final RTD is a comparison of R1 and R2 summed over all element-level divergence. It includes a normalization prefactor

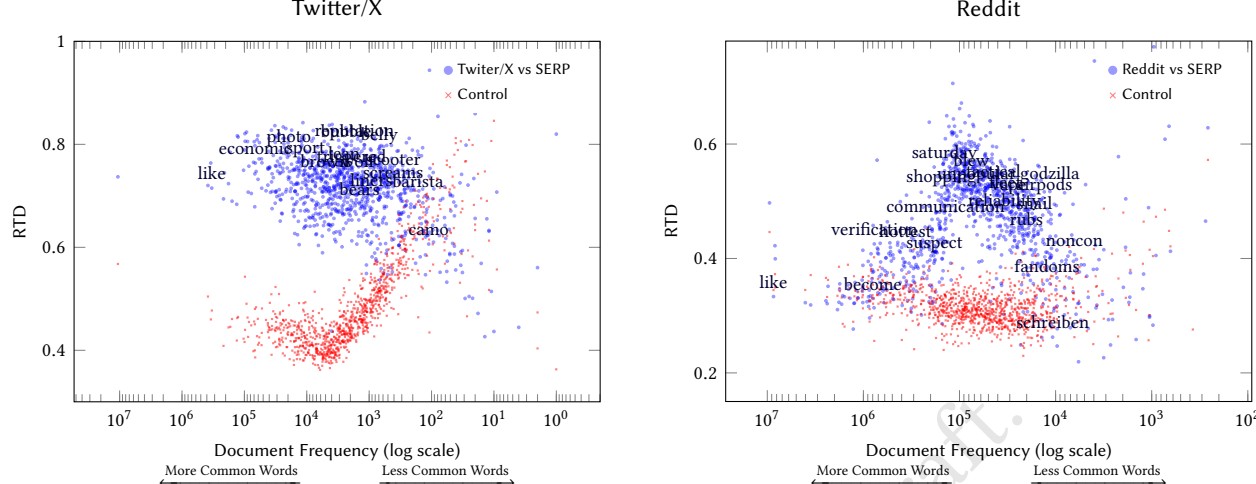

**Figure 4: Rank Turbulence Divergence (RTD) as a function of Document Frequency. Common terms and rare terms shared between SERP results diverge from the substantially nonsampled social media data.**

**Table 1: Rank Turbulence Divergence (RTD) between SERP results and the nonsampled social media data.**

| Site | **RTD** (SERP vs Social Media) |
|------|-------------------------------|
| Reddit | 0.47 |
| Twitter | 0.70 |

$N_{1,2;\alpha}$ and takes the following form:

$$RTD_\alpha^R(R1 \parallel R2) = \frac{1}{N_{1,2;\alpha}} \frac{\alpha+1}{\alpha} \sum_{\tau \in R_{1,2;\alpha}} \left| \frac{1}{[r_{\tau,1}]^\alpha} - \frac{1}{1/[r_{\tau,2}]^\alpha} \right|^{\frac{1}{\alpha+1}}$$

(3)

A lower score indicates little rank divergence; a higher score indicates a larger divergence. The mean RTD for SERP results from all 1000 keywords is listed in Tab. 1. Raw numbers are difficult to interpret on their own, but a control test performed in the next section found an RTD of about 0.30 on random comparisons of the same dataset. The RTD comparing SERP results for Reddit and Twitter/X were both dramatically higher than the control.

Overall, this domain-level analysis shows that SERP results are substantially different from each other. Next, our goal is to characterize the nature of this difference.

### 4.1 Term-level Analysis

Recall that the corpus-level RTD values presented above from Eq. 3 are the mean average of the rank divergence of the individual words from Eq. 2. This permits a token-level analysis to find the keywords that diverge the most within the dataset. We do this by capturing the output and the sign from Eq. 2 for each term in the posts/Tweets returned by SERP or returned by a keyword-query to the nonsampled social media data. Figure 3 shows the distribution of the term-level divergences (Eq. 2) and their mean (representing Eq. 3) for terms that have the highest mean rank divergence in favor of Google's SERP and in favor of the nonsampled social media data

from Twitter/X (on left) and Reddit (on right). In other words, the terms in red (*i.e.*, top subplots) are more likely to be returned from the Google's SERP compared to the nonsampled social media data, and vice versa.

On the nonsampled Twitter/X data we are far more likely to encounter hashtag-style terms, along with politically salient terms referencing then-President Trump, terms blaming China for the pandemic, and other terms with a generally negative sentiment. The results from SERP, in contrast, illustrate medical information (hospital, health, services, support), government offices (city, minister, socialsecurity, facility, district), and terms with a more-positive (or at least more-neutral) tone.

From the Reddit data we find that Reddit-specific terms like removed, comments, deleted, unsubscribe, etc are far more likely to appear on Reddit compared to SERP, which is reasonable, but also means that the search engine is more likely to hold-back these items from the search results. We also find that the nonsampled Reddit data is far more likely to have terms from American football (Romo, Mahomes, Bengals, Jags, refs), which was in its playoffs during the data collection period, political terms (Trump, Ukraine, McCarthy, neoliberal, republican, vote), and vulgarity.

### 4.2 Term-Frequency Analysis

Although a targeted social media analysis might intentionally develop a list of query words, like the COVID hashtags used to collect the Twitter / X data, recall that the keywords used in our token-level analysis were selected from a stratified sample of all terms ordered by their document frequency. Here, we ask whether there is a relationship between the frequency of a keyword (as measured by the document frequency) and its RTD.

We compute the RTD values comparing SERP results against the nonsampled social media data for each of the 1000 keywords. We compare these values against an RTD control set created by randomly assigning 5000 random posts from the nonsampled social media data for each keyword in R1 and R2.

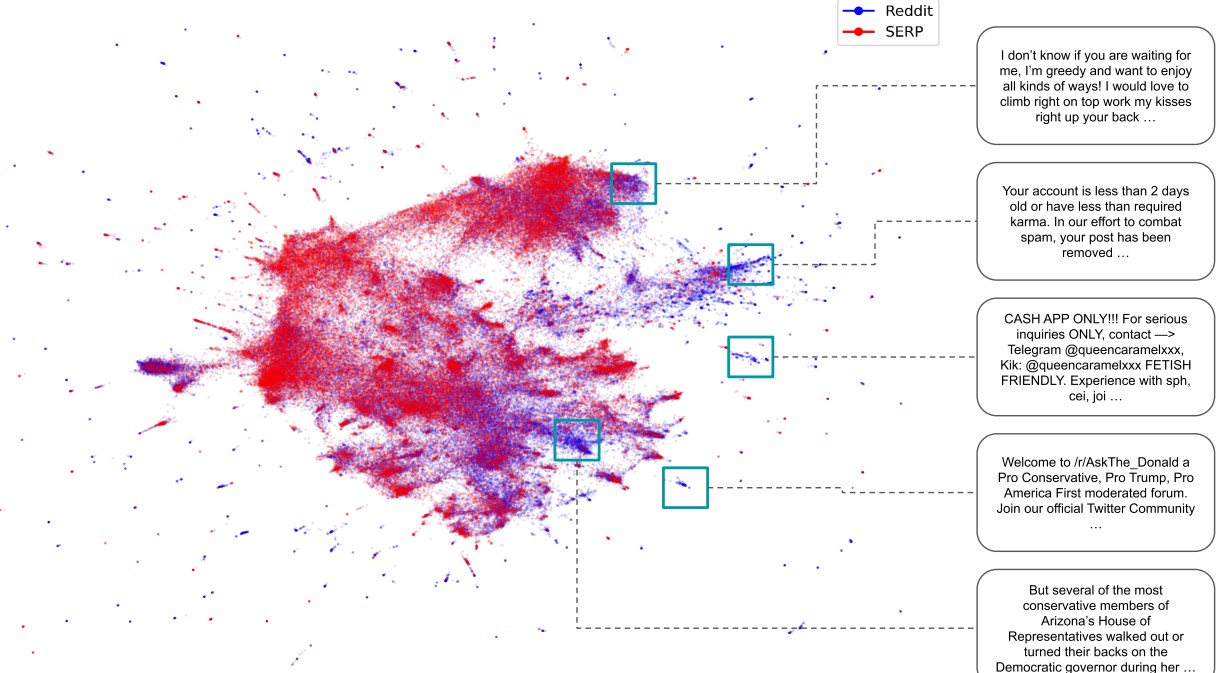

**Figure 5: Topical coverage map of the nonsampled Reddit data (blue) and posts returned from SERP (red). Clusters of blue show topical clusters that are found in the nonsampled Reddit data that are not returned by SERP. Examples of some of the clusters are listed on the right.**

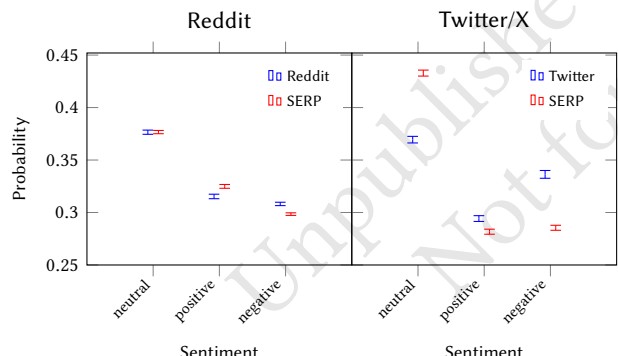

**Figure 6: Sentiment Probabilities and their 95% confidence intervals. SERP results were statistically more-positive or more-neutral than the nonsampled social media data.**

Compared to the control set, Fig. 4 shows that the RTD is consistently higher on the Twitter/X dataset. On the Reddit dataset, we find that the RTD starts out relatively low for the most common words, but then rises substantially for more-informative words with medium document frequencies, and then reverts to the control for the less common words.

Together, these results indicate that Google's SERP returns a highly skewed view of the underlying social media data, and that the difference is most pronounced in terms that are most informative.

## 5 SENTIMENT ANALYSIS

Social media has also been widely used to glean information regarding sentiment and emotional judgement regarding various topics [15]. Although we do not investigate any single-topic or event in the present work, we do make use of sentiment analysis tools to determine if SERP produces and bias in terms of sentiment or emotionally-salient language. We do this at the post-level and employ a sentiment analysis model called TimeLMs [17] based on the roberta [16] transformer architecture.

The findings from the term-level analysis also appeared to have a difference in the overall sentiment and emotional salience. Simply put, Google's SERP appeared to return social media posts that were much more positive compared to the nonsampled social media data. In this section, we perform a focused analysis of the differences in sentiment found in the corpora.

Sentiment analysis on social media has been used for decades to gauge the users' attitudes towards products, organizations, issues and their various facets [20]. Analysis of sentiment has become one of the widely researched areas in the recent times, and many large organizations have entire social media teams dedicated to managing their social media presence. In the Post-API era, it is important to understand if SERP provides an accurate characterization of the true distribution of sentiment found on social media [34].

Sentiment analysis tools can be deployed on various levels including sentence-level [9], document level [39], and aspect level [26] analysis. For this task we used a sentiment analysis model based on

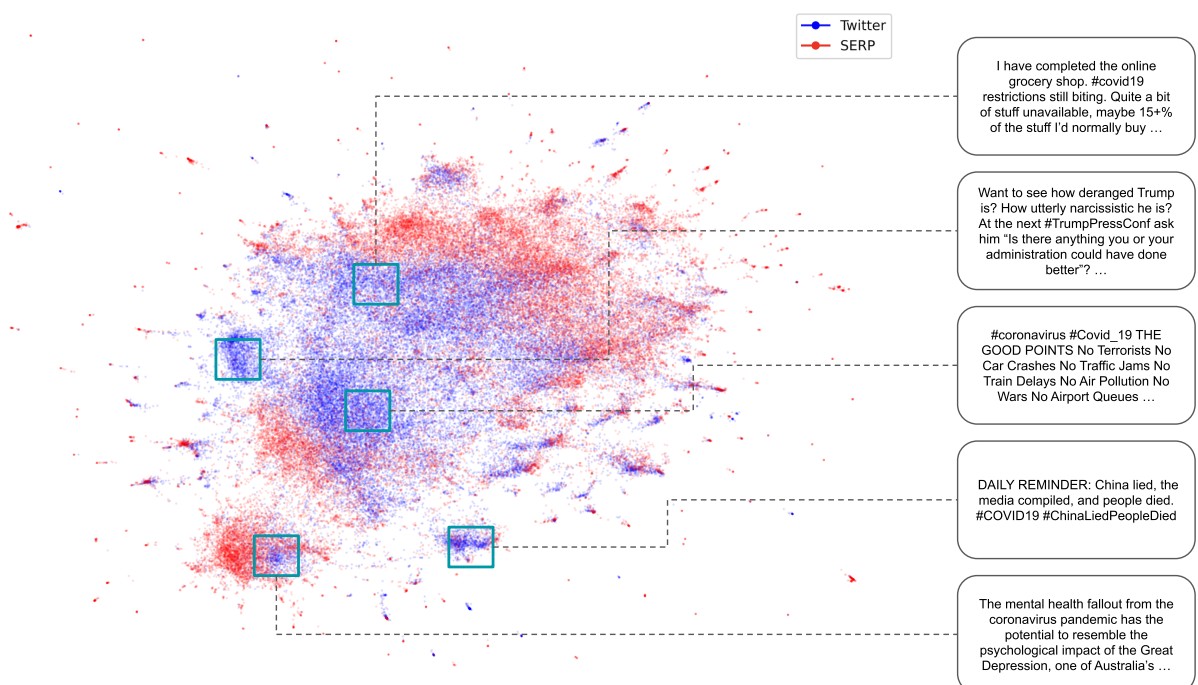

I have completed the online grocery shop. #covid19 restrictions still biting. Quite a bit of stuff unavailable, maybe 15+% of the stuff I'd normally buy …

Want to see how deranged Trump is? How utterly narcissistic he is? At the next #TrumpPressConf ask him "Is there anything you or your administration could have done better"? …

#coronavirus #Covid_19 THE GOOD POINTS No Terrorists No Car Crashes No Traffic Jams No Train Delays No Air Pollution No Wars No Airport Queues …

DAILY REMINDER: China lied, the media compiled, and people died. #COVID19 #ChinaLiedPeopleDied

The mental health fallout from the coronavirus pandemic has the potential to resemble the psychological impact of the Great Depression, one of Australia's …

**Figure 7: Topical coverage map of the nonsampled Twitter/X data (blue) and Tweets returned from SERP (red). Clusters of blue show topical clusters that are found in the nonsampled Twitter data that are not returned by SERP. Examples of some of the clusters are listed on the right.**

the Roberta transformer model [16] that was fine-tuned on Twitter data. The sentiment analysis model was applied to each Reddit post and Tweet; it returned a probability that each post was neutral, positive, or negative.

The mean-average of the sentiment probabilities and their 95% confidence intervals are plotted in Fig. 6. For Reddit, we find that the posts returned by SERP were statistically more-positive than the nonsampled Reddit data ($p < 0.05$) and vice versa. The shift towards more-positive was also reflected on Twitter/X, where we see large, statistically significant increases in the number of neutral Tweets and statistically significant decreases in the number of negative Tweets returned by SERP compared to the nonsampled data.

## 6 GAPS IN TOPICAL COVERAGE

Our final analysis investigates the topical coverage of the data. If SERP returned an unbiased sample of the underlying social media data, then we would expect that the topical coverage of the results returned from SERP would have a similar topical distribution of the nonsampled social media data.

Topical analysis on text data is a well-understood and deeply investigated mode of analysis starting from latent semantic indexing (LSI) and latent Dirichlet allocation (LDA) models in the past [3, 4] to learned vector representation models such as word2vec [22] and GLOVE [29]. But these term-based models do not provide a contextual understanding of sentence-level semantics found in social

media posts. Sentence Transformers, on the other hand, provide contextual whole-sentence embeddings of entire sentences or paragraphs [32]. Therefore, sentence transformers are a powerful tool for tasks such as sentence similarity, clustering, and retrieval.

In order to compare the topical coverage of SERP results against the nonsampled social media data, we used a pretrained multilingual sentence transformer model called `paraphrase-multilingual-mpnet-base-v2` [32] to encode each social media post and Tweet, from both the nonsampled social media data and from SERP, into a 768-dimensional vector space. Because this pretrained transformer model was not fine-tuned on any of the datasets, it will encode the posts from all of the datasets into the same high-dimensional semantic space. We then used UMAP to project the high-dimensional embeddings into a shared two-dimensional projection [19].

The resulting plots are illustrated in Figs. 5 and 7 for Reddit and Twitter/X, respectively. A complete, interactive visualization of these plots is available here (viewer discretion is advised). In both cases, the nonsampled social media data from Reddit and Twitter/X is plotted in blue; the results from SERP are plotted in Red and always in front of the nonsampled social media plots. Because the results from SERP are a subset of the nonsampled social media data, the points visible in red usually elide and therefore match the same post from the nonsampled social media data. As a result, the points visible in blue indicate gaps in topical coverage in SERP results.

Topical gaps in Reddit coverage are illustrated as blue points in Fig. 5. We identified several topical-areas where Reddit data was not covered by results from SERP; five of these areas are selected and a representative sample of the social media post. One exception is in the top-most cluster, which contained mostly pornographic posts; this particular example was deliberately chosen to sanitize the illustration from highly graphic and sexually explicit language, which make up the majority of this cluster. Overall we find that SERP generally censors Reddit posts that are pornographic, spam, highly-political, and contain moderation messages.

We find similar coverage gaps on Twitter/X illustrated in Fig. 7. Several topical gaps are evident; we focus on five clusters with representative Tweets illustrated on the right. Perhaps the tightest cluster in this illustration focus on the hashtag #ChinaLiedPeopleDied; another focuses on negative political aspects of then-President Trump. Generally, the coverage gaps appear to highly align with the sentiment analysis from the previous section and can be broadly characterized as focusing on negative content, while SERP results tend to focus on healthcare-related content.

## 7 DISCUSSION

The results of this study, overall four dimensions of analysis, clearly show that SERP results are highly biased samples of the social media data. Although this is not unexpected, the nature of these differences were surprising. This analysis is an early step in understanding the tradeoffs that result in the use of SERP results as a replacement for API access.

We summarize the results as follows: (1) We found that SERP results return posts from Twitter/X users that have a dramatically larger following than the average Twitter user; likewise, for Reddit we find that that SERP results return posts that have a dramatically higher score than the average Reddit post. Unexpectedly, we did not find any correlation between user popularity or post score and its rank in the SERP results. (2) Token-level analysis found a substantive difference in the likelihood of various terms appearing in posts returned by SERP. SERP results appeared to be less political, less vulgar, and, on the COVID-oriented Twitter/X dataset, far more likely to mention social and health services. (3) The token-level analysis appeared to show that SERP results were generally more positive than the nonsampled social data. Indeed a full-scale sentiment analysis showed that SERP results tended to be statistically more-positive than the average social media post. (4) Finally, maps of topical coverage indicated vast swaths of the semantic space were missing from the SERP results. Further investigation found that pornographic, vulgar, political, and spam posts were largely absent from SERP results.

### 7.1 Cost Analysis

At present, nearly every social media platform either charges for API access, severely limits access, or does not provide API access at all. The nonsampled data used in the present work was collected prior to API access being put into place. Nevertheless, we wish to provide an analysis of what it would cost to collect the nonsampled data at present rates. For this we considered three sources: Reddit API, Twitter API, and ScaleSERP API. The Reddit API, which charges 24 cents per 1,000 API requests, would cost 240 USD to obtain 1

million posts. The Twitter API comes at a significantly higher price, with a cost of 5,000 USD for 1 million posts per month. ScaleSERP uses a slightly different payment model; it is important to note that ScaleSERP generally provides a maximum of 100 results (if available) per call. As a result, approximately 10,000 queries are required to retrieve 1 million posts. The cost for 10,000 API requests from the ScaleSERP is 59 USD. Using SERP is clearly a cost-conscious decision, but does come with a price of a highly biased sample.

### 7.2 Threats to Validity

This present work is not without limitations. To begin, the initial assumption of our analysis is that the data gathered from Reddit and Twitter/X are (almost) complete. Although most social media posts grow stale after a few days [12], any user may comment, retweet, like, or upvote any post at anytime; as a result this data may not account for social activity that occurred after the time of capture. In a similar vein, although the the Twitter/X dataset is (almost) complete for the 7 COVID hashtags, these hashtags certainly do not comprehensively encompass all of the discussions surrounding COVID.

Another notable limitation stems from the non-deterministic nature of SERP results. The data retrieved from SERP may or may not appear at the same rank (or at all) when re-queried. This could impact our analysis, particularly the rank-based correlation results in the analysis of popularity. Given that we query SERP with 1,000 keywords, the samples collected should still provide valuable insights for our study.

### 7.3 Conclusions

Taken together, these findings collectively point to a large bias in SERP results. This raises the question on the validity of any research that is performed with data collected from SERP only; however, we currently know of none. Overall, we conclude that SERP is not a viable alternative to direct access to social media data.

Future research that heavily relies on SERP results may provide value, but it is important that these future works are cognizant of the limitations and biases in SERP results and are careful not to make conclusions that do not rely on an unbiased sample of social media data. However, it is also important to highlight the cases where SERP results can serve as a trustworthy data source. For example, studies which study search engines can make natural use of SERP results; likewise, SERP may be used as a seed set for additional analysis.

Although the present work answers many questions, it raises others as well. We are additionally interested in the differences between SERP results and Web-scraped results. For example, it could be the case that SERP results are actually a unbiased sample of the results that social media platforms provide.

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
