# OpenReview forum: "Navigating the Post-API Dilemma: Search Engine Results Pages Present a Biased View of Social Media Data"
_ACM.org/TheWebConf/2024/Conference — TheWebConf24 Oral_

### Official Review · Reviewer_7BHH · 2023-10-27

**Novelty:** 5
**Technical Quality:** 5

**Review:**

The paper investigates the use of search engines to obtain datasets from social media platforms like Twitter and Reddit. This paper is extremely relevant and useful nowadays, especially after Twitter and Reddit made changes to their APIs that make getting large-scale datasets a difficult task. To the best of my knowledge, this paper is novel, and I have not seen any paper that attempts to study this phenomenon. Despite the fact that the paper has a negative outcome (i.e., it concludes that the use of search engines for obtaining datasets from social media platforms is biased and incomplete) indicating that search engines can not replace the platforms’ APIs, I still believe that the paper has merit and can inform other researchers that seek alternative methods in collecting datasets.
 Overall the paper has the following notable strengths:
1. Important and timely work that can have a big impact on the Computational Social Science community
2. Use of large-scale datasets across multiple platforms (Twitter and Reddit).
3. Analysis across multiple aspects, including popularity, topics, and sentiment.
4. Employed methods are well-suited for the intended analysis.

At the same time, I believe that the paper, naturally, has some weaknesses (I elaborate on these weaknesses below):
1. The sentiment analysis method is not validated across the two platforms
2. Unclear if the differences are due to pagination on Google or due to the differences in content moderation policies
3. The Twitter dataset is only related to COVID-19, which is limiting, and it is unclear how this might affect the results (i.e., likely Google moderated COVID-19-related information)
4. The analysis of the coverage is shallow and does not provide a complete picture. For instance, there is no subreddit-level analysis that will be extremely useful to researchers to figure out if specific subreddits are not available on search engines at all or if there are substantial coverage differences across subreddits.

One of my concerns with the paper is that it uses a sentiment analysis tool that is trained on Twitter data, however, the paper uses it on Twitter and Reddit datasets that are substantially different. So it’s unclear if the presented differences in the sentiment are actually because of the sentiment or the difference in the performance of the classifier across the two platforms. I suggest to the authors to consider validating the performance of the sentiment analysis across the two platforms by performing a small-scale validation on a random sample of posts from Reddit and Twitter to quantify the performance across the two platforms.

Second, the paper finds that the data from SERP are incomplete, however it is unclear whether this lack of coverage is due to the pagination and ranking of the results on Google or due to the fact that Google has a substantially different moderation policy compared to the other platforms. I would have liked to see a deeper analysis on these issues.

Third, an important limitation of this paper is that the Twitter dataset is only related to COVID-19. Overall, it’s unclear how this factor affects the presented results, especially because it’s likely that Google moderated content related to COVID-19 because of the severity and impact of the issue in society. I suggest to the authors to discuss this limitation and how it can affect the results and probably consider expanding their analysis with another dataset from Twitter that is not related to COVID-19.

Finally, I believe that the presented analysis of Reddit is very shallow and lacks a subreddit-specific analysis. After reading this paper, I am wondering whether there are substantial coverage differences across subreddits (i.e., are some subreddits completely absent while others are complete). Also, it’s unclear if researchers can use search engines to obtain data from specific subreddits rather than Reddit as a whole. Overall, I believe that the paper missed a great opportunity to make a deeper analysis on Reddit that will provide important information to the research community.

Taken all together, I believe that this paper makes a nice contribution, however, there are also some important limitations and weaknesses with the presented analysis. Due to this, I would classify this submission as “accept if room.”

**Questions:**

1. In the case of Reddit, have you observed specific subreddits not included at all in the results returned from SERP?
2. What is the performance of the sentiment analysis tool in your datasets?
3. How are the Twitter results affected because all tweets are related to COVID-19, content that might get moderated by Google?

**Ethics Review Description:**

No ethical issues

**Reviewer Confidence:**

3: The reviewer is confident but not certain that the evaluation is correct

**Scope:**

4: The work is relevant to the Web and to the track, and is of broad interest to the community

---

### Official Review · Reviewer_S9Af · 2023-11-20

**Novelty:** 6
**Technical Quality:** 5

**Review:**

## Summary
This study explores using results from search engines (SERP) as a replacement for the (now) virtually inaccessible Twitter and Reddit API. The studies assess the differences between the data obtained from SERP and the Twitter, Reddit APIs. The summary of their findings is well presented in the discussion section.

Pros:
- This is important work and relatively untouched territory.
- Well written (though some clarification can help better the paper. See Questions section)
- Formally introduces and presents a reasonable analysis of an alternative (SERP) to collecting data Twitter/Reddit API. The analysis circles how different/similar data from SERP is.

Cons:
- Completeness of the dataset which the search engine results were compared against.
- A strong justification for the metric used to define popularity in Reddit and Twitter is missing. Given the final results are dependent on this metric, it seems essential to test the robustness of the findings against other metrics of popularity that can be defined and justified.
- It is unclear if a different sample of keywords used to query SERP would have changed the conclusions of this work.

Overall, I find this to be important and meaningful work.
I thank the authors for choosing to work on this problem.

However, I suggest and request some clarifications summarised in the "Questions" box below.

**Questions:**

- Elaborate on the Tokenisation scheme.
(a) Did you just split on spaces and remove alphanumerics? Or did you use sentence-piece or something like that?
(b) Did you remove stop words? Why or why is this not important to do?
(c) Comment on how this step may better the stratified sampling of keywords.

- Elaborate on stratified sampling: Assuming you grouped tokens (made the strata) based on document frequency, what are the widths of these bins? In total, were there just 1000 bins and you sampled one word from each bin?

- Equation 3 has an extra 1 in the denominator (?).

- I would rename the section from ``token-based comparison`` to ``keyword-based comparison`` because what it is essentially doing (according to line 502) is comparing the ranks of the 1000 keywords used to query SERP with their rank in the reddit/twitter data.

- I think R1 and R2 are used differently in line 580 compared to their usage before (e.g., in equation 3 or line 443).

- How do you know “This dataset is considered to be a nearly-complete set of tweets” Does the paper say so? Perhaps hashtag search does that. Please clarify this.

- Given that the sample you are comparing with plays such a major role, it will be useful to offer a short summary of how this comparison sample was curated (especially in the case of X)

- The Twitter query to SERP was of the form: site:twitter.com {hashtag} {keyword} for each keyword. Clarify if this is an AND search or an OR search.

- It is unclear if the sample of keywords and hashtags used in the case of Twitter were based on stratified sampling like in the case of Reddit. It seems like hashtags were sampled randomly. Why not stratified sampling? Please clarify.

- Is the total number of comments or upvotes available in the SERP data? Maybe better to use that as a popularity metric than a score (which is the number of upvotes minus downvotes…a highly popular post may have a 0 score). Similarly, the number of retweets instead of the number of followers might be a better judge of the popularity of the post. The latter judges individual popularity. In other words, how robust are your findings to different metrics of popularity?

- "The results gathered from SERP were surprisingly small. In total SERP gathered 1,296,958 results from Reddit 318 and 70,018 tweets from Twitter/X."
Was this because of some default parameters in the SERP API since the previous paragraph mentions that the default parameters were used...

- In line 461, it is mentioned we use “ alpha = ⅓ empirically”. What do you mean by “empirically?” Elaborate how this was empirically obtained.

- In line 458 you say you take care to preserve the sign but equations 1,2, and 3 indicate that the absolute value (shown by the modulus operator) of element-wise RTD is taken. This is confusing. Please clarify

- While an RTD of 0.3 on random comparisons was found, are the numbers shown in Table 1 "significantly" different from this? Please clarify the usage of significance here. Were any statistical tests performed to judge this?

- Is there a benefit to using RTD for term-level analysis to produce (say) figure 3 over something like log-odds with a Dirichlet prior?

- In sec 4.1, I suggest renaming ``term`` to ``token`` to be consistent with the previous sections
how is a distribution of term level divergences obtained? From my understanding, for the word "support," you calculate its rank in the SERP data and that in Twitter and evaluate the RTD for this based on equation 2. This will result in a single number. Where then is a distribution for the RTDs of "support" coming from? Perhaps I’m missing something/further clarification is required.

- In line 632, how do you know that the terms with medium document frequency are the "more-informative" words? What does "more informative" mean?

- What is the value of N1,2;alpha in equation 3?

- What statistical test was used in line 735? Please clarify. I also see a decrease in positive sentiment tweets in Fig 6 Twitter (not sure if this is significant or not). How then can you say in line 736 that there is a shift to more positive on Twitter simply cause there is an increase in neutral and a decrease in negative?

- Since sentiment analysis is done before topic, I suggest the figure for it should come before the one for topic.

- Regarding the topical analysis, given that SERP sample was created based on a list of 1000 keywords from stratified sampled based on the document frequency of tokens in the non-sampled dataset, it is expected that this misses certain topics. Of course one can think of sampling 1000 keywords such that it covers a topical distribution of the non-sampled data. So concluding that SERP does not offer the expected coverage of topics is not the right conclusion according to me unless the authors convince us (the readers) otherwise.

- A major concern of this work is how the keyword sample was created to query SERPA. A different sample of keywords would have changed the conclusions of this work. This should be perhaps mentioned in section 7.2. I would soften the language in the conclusion. For example, in line 904 “is not a viable alternative” to “may not be a viable alternative”

-  I need help understanding line 918. What do you mean they are actually an unbiased sample?
Also, perhaps it is useful to discuss the possibility that results from other search engine APIs might have other kinds of biases if not for the ones found in this study and is an interesting direction for future work.

**Reviewer Confidence:**

4: The reviewer is certain that the evaluation is correct and very familiar with the relevant literature

**Scope:**

4: The work is relevant to the Web and to the track, and is of broad interest to the community

---

### Official Review · Reviewer_uMAQ · 2023-11-27

**Novelty:** 6
**Technical Quality:** 5

**Review:**

The paper discusses the possibilities of using data provided by SEPR as a potential replacement for data from platform API which are no longer available. It concludes that there are substantial distinctions between the samples provided by SEPR and by APIs.

Pros: 1) The topic is highly relevant for web research and addresses the  problem in a novel way; 2) The findings generated through the research design are interesting and can inform future studies' design regarding the potential of using SEPR data; 3) The methodology is sound and offers a comprehensive comparison of samples provided by SEPR and APIs; 4) The discussion offers a well-thought summary of the study, including some of its limitations (albeit not all).

Cons: 1) My major concern relates the basic premise of the study - i.e. the comparison between SEPR and API data. As far as I see, the study does not report when SEPR data was acquired, but it compares it with API data (for Twitter) coming from 2020; for Reddit data, it is not clear when exactly API-based dataset was established. Under these circumstances, my question is: to what degree differences in SEPR data are attributed to the different time points when data were collected - could not some data from Twitter been deleted (and, thus, disappear from SEPR data)? It can be a rational explanation of the differences in terms of thematic composition between Twitter API data and Twitter SEPR data; 2) My second concern relates to the initial premise of the article: i.e. that SEPR data can be viewed as a replacement for API data (unless SERP data is biased). However, I am not fully convinced this premise is valid concerning many limitations of SEs indexing approaches in the context of social media data. I.e. how realistic at all to rely on SERP data for studying social media platforms? Would not be web archives (like Internet Wayback Machine) be a more feasible alternative in the first place? 3) My third concern relates to the lack of related research section. I think it would be useful to add a bit more discussion of the existing discussions of sampling issues / limitations of different approaches for social media data aquisition (including digital archiving).

**Questions:**

How much of observed differences between API and SEPR datasets can be attributed to the time-based differences between the datasets?

Is it realistic at all to use SEPR data for sampling social media data? Are there any studies attempting it (also considering the non-deterministic nature of SE outputs noted by the authors)?

**Reviewer Confidence:**

4: The reviewer is certain that the evaluation is correct and very familiar with the relevant literature

**Scope:**

4: The work is relevant to the Web and to the track, and is of broad interest to the community

---

### Decision · Program_Chairs · 2024-01-22

**Decision:**

Accept (Oral)

**Comment:**

All reviewers agree about the merits of this paper. I also appreciate the efforts put forth by the authors to clarify many of the points raised in the reviews. I recommend acceptance of this paper, contingent on addressing the asks mentioned in the reviews. Here is a non-exhaustive list of the asks:
 - Reviewer uMAQ was not convinced about the following point made in the paper: "Data may have been deleted from X/Twitter, but the Google/SERP indexer...." and they ask for additional clarity in the revised version of the paper.
 - Reviewer S9Af asks for multiple clarifications points to be included in the final revised version of the paper and I hope the authors will be able to make those updates
 - Reviewer 7BHH asks authors to be upfront about the limitations of this work. Please expand the limitation section in the revised version.

 Finally, I suggest that the authors pay close attention to each point mentioned by the reviewer, reflect on their plan to address those and then update/revise the paper accordingly. I think this will make the work stronger.